# Sour Beer with *Lacticaseibacillus paracasei* subsp. *paracasei* F19: Feasibility and Influence of Supplementation with *Spondias mombin* L. Juice and/or By-Product

**DOI:** 10.3390/foods11244068

**Published:** 2022-12-16

**Authors:** Ana Beatriz Praia, Marcos Edgar Herkenhoff, Oliver Broedel, Marcus Frohme, Susana Marta Isay Saad

**Affiliations:** 1Department of Biochemical and Pharmaceutical Technology, School of Pharmaceutical Sciences, University of São Paulo (USP), Av. Professor Lineu Prestes, 580, São Paulo 05508-000, SP, Brazil; 2Food Research Center FoRC, University of São Paulo (USP), Av. Professor Lineu Prestes, 580, São Paulo 05508-000, SP, Brazil; 3Division Molecular Biotechnology and Functional Genomics, Technical University of Applied Sciences Wildau, 15745 Wildau, Germany

**Keywords:** sour beer, probiotics, co-fermentation, fruit by-product

## Abstract

This study aimed to evaluate the probiotic strain *Lacticaseibacillus* (L.) *paracasei* subsp. paracasei F19 (F19) with the yeast *Saccharomyces cerevisiae* US-05 (US-05), using *Spondias mombin* L. (‘taperebá’ or ‘cajá’) juice and by-product, in four sour-type beer formulations: control, with bagasse, juice, and juice and bagasse. The viability of F19 was evaluated by pour-plating and PMA-qPCR. Fermentability, in addition to physicochemical and sensory parameters, and aroma and flavor, were evaluated during brewery by using Headspace Solid-Phase Microextraction (HS-SPME) coupled with gas chromatography–mass spectrometry (GC–MS). F19 was successful in fermenting bagasse in a MRS medium (9.28 log CFU/mL in 24 h) but had a low viability in hopped wort, growing better in formulations without bagasse or juice. No difference between formulations was observed regarding sensory acceptability, and the HS-SPME/GC-MS revealed different flavors and aroma compounds. In conclusion, the production of a potential probiotic sour beer with F19 and US-05 is feasible regarding probiotic viability. However, *S. mombin*, as juice or bagasse, threatened probiotic survival. Different flavors and aroma compounds were detected, whereas no difference between formulations was found regarding sensory acceptability. The moderate alcohol content achieved is important for bacterial survival and for the development of a probiotic beer with health claims.

## 1. Introduction

Beer is one of the most consumed beverages in the world and moves a billionaire market. Despite the large industries representing a domain in this market, craft breweries have been growing lately and gaining more space by investing in the development of new fermentation processes and flavors, with the addition of different ingredients, such as fruits, herbs, spices and other ingredients [1,2].

Malt, hops, and water are the basic ingredients for beer brewing, listed in the German Beer Purity Law of 1516, created to avoid using grains such as wheat or rye in beverage production. Later, after discovering that they were responsible for the fermentation of beer, yeasts were also included in this ingredients list [3]. Over time, the concept of beer has changed, and different types and beer styles have emerged, either due to changes in the brewing, temperature, geographic location, type of microorganisms or the addition of other ingredients, contributing to a beer flavor diversity. Nevertheless, the beer production is still based on a fermentation process of boiling malt with hops [2,4].

Sour beer has stood out in recent decades for its unique flavor and for not being a style restricted to a particular production process, origin or ingredient used, as it is characterized by acidity resulting from the high concentration of organic acids and low pH (3.0 to 3.9). Consequently, in most cases, it results from the fermentation of lactic acid bacteria (LAB), such as *Lactobacillus* spp. [4]. Therefore, there are many alternatives to be explored in the fermentation process to achieve new production techniques and flavors for this beer style [5,6].

There are many variations of sour-type beers in several countries, with Catharina Sour being the Brazilian representative [7]. Catharina Sour is a style of beer that has fruit or fruit juice added in its fermentation. In Brazil and in the world, there are a multitude of fruits available to be used in the fermentation of this style of beer. 

Known as Taperebá in northern Brazil, and as Cajá in the Northeast, *Spondias mombin* L. is an oval-shaped, small, yellowish-colored fruit that contains, in its center, a seed that occupies a considerable part of the fruit, surrounded by a pulp layer and a thin peel. It is known for being a good source of vitamin A, antioxidants, and carotenoids, in addition to having a slightly sour taste and a pleasant aroma [8,9]. The *S. mombin* frozen pulp is widely consumed and has a well-defined market and is used as an ingredient for the preparation of foods such as juices, ice cream, sweets, popsicles, and mousses [8]. However, the by-product of the pulping process, which has nutritional potential and might be used in the preparation of several products is, in general, discarded [10].

Fruit by-products consist of unused raw materials rich in nutrients, bioactive compounds and are excellent fermentation substrates, as they contain sugars, vitamins and minerals, and fiber sources, which contribute to the development of beneficial bacteria in the intestinal tract [10,11,12]. It is estimated that about a third of the food produced worldwide is wasted per year and the agricultural sector is responsible for most of this amount [13]. Faced with this scenario, measures aimed at reusing viable and nutritious raw materials which, in general, are wasted, deserve to be highlighted, since they may contribute to both the economy and the environment.

LAB may have an effect both on human health, improving the intestinal tract and the immune system [14], and in food technology, acting in beer production, either through malt acidification, which results in protection against microbial spoilage, or by the definition of sensory attributes [15]. At the same time, the presence of bacteria in beer is often seen as a deteriorating agent, and the beer itself has limiting factors which avoid bacterial development, such as hops and alcohol production [4,16].

Therefore, this study aimed to evaluate the feasibility of the application of the probiotic bacterial strain *Lacticaseibacillus* (L.) *paracasei* subsp. *paracasei* F19 [17] in sequential fermentation with the yeast *Saccharomyces cerevisiae* US-05 in a sour beer production, with the addition of *Spondias mombin* bagasse and/or juice; it also aimed to evaluate the physical–chemical properties, sensorial parameters, and the aroma and flavor profile of the resulting beer formulations.

## 2. Materials and Methods

### 2.1. Ingredients and Microorganisms Employed

The probiotic culture *Lacticaseibacillus* (L.) *paracasei* subsp. *paracasei* F19 used in this study was supplied by Chr. Hansen (Hørsholm, Denmark), and the dry yeast *Saccharomyces cerevisiae* US-05 (SafAle™ US-05) was supplied by Fermentis (Barœul, France).

Four replicates of the *L. paracasei* F19 strain, kept frozen at −80 °C in glycerol (800 µL of the probiotic strain in 200 µL of glycerol), were thawed and cultivated twice consecutively in MRS broth at 37 °C for 24 h, followed by centrifugation at 10,000× *g* for 10 min, washed with sterile saline solution (0.85 g NaCl/100 mL) and homogenized using a stirrer. This process was repeated three times in sequence and the supernatant was discarded. The pellet obtained was re-suspended and used to inoculate the beer formulations at an approximate concentration of 8 log CFU/mL.

The Cajá’s (*Spondias mombin* L.) by-product (fruit peel and pulp residues, without seeds) was donated by a frozen fruit pulp industry located in Natal, Rio Grande do Norte, Brazil. It was transported to the Laboratory of Bioactive Compounds in Food (LABTA) of the Faculty of Food Engineering at the Federal University of Rio Grande do Norte (UFRN), where it was mixed in plastic containers and subdivided into plastic food bags, in portions of 500 g each, and stored at −22 °C for further dehydration. The by-product portions were then transported frozen and thermally isolated to the University of São Paulo (USP), where the following procedures were carried out: defrosting at 4 ± 2 °C for 48h, followed by whitening in boiling water for a few minutes and cooling in an ice bath. Next, they were dried in an air circulation oven at 60 °C for 24 h, with subsequent storage in sterile plastic food bags suitable for food. For the beer production, the dried *S. mombin* by-product was milled and sieved to obtain some flour with an approximate particle size of ≤ 0.42 mm.

The *S. mombin* juice used was obtained from fruit frozen pulp (DeMarchi, Jundiaí, São Paulo, Brazil) purchased in a local market in São Paulo—SP, Brazil. The juice was prepared on a laboratory scale, following the pulp manufacturer’s recommendations, at a 1:2 ratio (pulp: water), reaching 4 °Brix and a pH of 2.7. The juice used was composed only of fruit pulp, without any added sugar or any type of food preservative and/or additive. Before being added to the beverages, it was submitted to a pasteurization process for 30 min at 60 °C, with subsequent cooling in an ice bath and with the aid of a mechanical stirrer (Mechanical stirrer 713D, Fisatom, São Paulo, Brazil) until reaching a temperature of 37 °C.

The *S. mombin* bagasse and juice used were characterized by soluble solids, pH, and total acidity content according to the methodology described in 2.7. For the beer production, the following ingredients were used to produce 5 L of beer: 730 g of Chateau Pilsen malt (Castle Malting, Beloeil, Belgium) and 730 g of Chateau wheat malt (Castle Malting, Beloeil, Belgium), 15 L of mineral water (pH 6.49) (Appendix A), and 4.2 g of Saaz hops (Saaz, Czech Republic). The recipe was designed by the BeerSmith^TM^ software v 3.2.7 (BeerSmith LLC, Clifton, VA, USA) according to the Beer Judge Certification Program (BJCP) [7] requirements for the Catharina Sour style.

### 2.2. Experimental Design

For this study, four types of beer formulations were tested: FC, FB, FJ, and FBJ. The four formulations were based on the same production process; the last step, where extra ingredients were added, was responsible for the differentiation between the formulations. FC was the control formulation, without any extra ingredients. The formulations and their respective ingredients are described in Table 1.

For the beer formulation, the *S. mombin* flour and the pasteurized juice were used. All equipment, containers, and utensils used were previously sanitized and sterilized in an autoclave at 121 °C for 15 min. The flowchart with all the production stages of the sour beer, with the addition of bagasse and/or *S. mombin* juice, is described in Figure 1.

### 2.3. Brewing

The first step of the brewing process was the wort preparation, where the Pilsen and the wheat malt were added to 4 L of water at 50 °C, heated to 72 °C for 60 min, and then to 76 °C for 10 min (Figure 1). The temperature was monitored throughout this process with a digital thermometer. After the mashing step, the wort was filtered and boiled for 60 min, and the rest of the water was added to the hops. Finally, after the boiling stage, the wort was divided into 4 glass bottles of 1 L and immediately cooled in an ice bath until reaching a temperature of 37 °C. The chemical composition of the water used is described in Appendix A.

After cooling, the filtered wort was inoculated with the probiotic strain *Lacticaseibacillus* (L.) *paracasei* subsp. *paracasei* F19 (in reference to the last taxonomy classification of the previous *Lactobacillus paracasei* subsp. *paracasei* F19 strain) [18] and incubated for 24 h at 37 °C (first fermentation). After 24 h of fermentation, the *S. mombin*’s bagasse and juice were added according to the previously defined formulations (Table 1): FB (addition of 10 g of bagasse); FJ (addition of 100 mL of juice); FBJ (addition of 10 g of bagasse and 100 mL of juice); and FC (control). After adding the extra ingredients, all formulations were inoculated with the yeast *Saccharomyces cerevisiae* strain SafAle™ US-05 in a concentration of 5 log CFU/mL and stored at 18 °C for 8 days for the secondary fermentation. The yeast rehydration was performed according to the manufacturer’s information (Fermentis, Barœul, France) and 3.8 g of yeast was sprinkled in 38 mL of sterile water and left to rest for 15 to 30 min before being added to the beers.

At the end of fermentation, the maturation stage began, with a gradual decrease in temperature: 13 °C for 2 days, 4 °C for 3 days, and 0 °C for 4 days. At the end of this process, the beers were bottled and priming proceeded; inverted sugar (fructose) was used (Lamas Brew, Campinas, São Paulo, Brazil) and added in the amount of 9.5 g/L. it was used to begin the final stage of production (carbonation), after which the beers were kept at 24 °C for 12 days.

### 2.4. Fermentability Assay

To evaluate the ability of the strain to ferment *S. mombin* by-products (as described in 2.1), an in vitro test was carried out using different types of medium: a MRS broth modified with the addition of phenol red, according to Vieira et al. [12], and beer wort with and without the addition of hops. The fermentation capacity of *L. paracasei* F19 was evaluated by comparing the media with and without 1% of irradiated *S. mombin* by-product flour. Thus, six mediums were evaluated: MVF (MRS broth modified with the addition of phenol red); MVF-BP (MRS broth modified with the addition of phenol red + *S. mombin* by-product flour); UNW (unhoped wort); UNW-BP (unhoped wort + *S. mombin* by-product flour); HPW (hopped wort); and HPW-PB (hopped wort + *S. mombin* by-product flour). A control medium without any inoculation of microorganisms was also tested, to confirm that there was no contamination in the by-product.

For the assay, the strain *L. paracasei* F19, in an initial concentration between 5 and 6 log CFU/mL, was inoculated in 5 mL modified MRS broth supplemented with *S. mombin* by-product and 5 mL of wort with and without hops. Next, the inoculated broths were incubated anaerobically (Anaerogen™ Anaerobic System, Oxoid, Thermo Fisher, Basingstoke, Hampshire, UK) at 37 °C for 48 h, and counting was performed at times 0 h, 24 h, and 48 h on DeMan–Rogosa–Sharpe agar (MRS agar, Oxoid, Thermo Fisher, Basingstoke, Hampshire, UK), using the drop plate technique [19].

### 2.5. Viability of Microorganisms by the Plate-Count Method

The viability of the microorganisms was determined by the pour-plating technique at the bacterial inoculation, 24 h after the first fermentation, in the critical periods when there was a change in temperature or production stage (fermentation, maturation, and carbonation), and within 30 days of storage under refrigeration at 4 °C (Table 2).

For the probiotic counts, 1 mL samples of the four beer formulations were diluted 1:10 in tubes with sterile saline solution (0.85 g NaCl/100 mL) as diluent. Serial dilutions were prepared, followed by 1 mL seeding of the dilutions in acidified MRS agar, in duplicates using the pour-plate technique, and then incubated in anaerobic conditions (AnaeroGen™ Anaerobic System, Oxoid) at 37 °C for 48 h, for the subsequent bacterial counts. The same method was used in all steps of the conventional microbiological analysis.

### 2.6. Viability of Microorganisms by PMA-qPCR

For the extraction of DNA, samples were previously treated with PMA (propidium monoazide) (Biotium, Hayward, CA, USA) to avoid amplification of the dead cell’s DNA. Before treatment with PMA, cell pellets were collected by centrifugation (9000 g/10 min at 4 °C), washed twice with Tris EDTA buffer (10 mM Tris-HCL, 1 mM EDTA, pH 8), and re-suspended in 499 µL of buffer PBS (phosphate buffered saline). Next, the treated samples were incubated in the dark for PMA activation for 5 min and placed in a Glo Plate™ Blue LED Illuminator (Biotium), applying LED light for 15 min for PMA deactivation [20]. DNA extractions were carried out using a MagMAX^TM^ kit (Thermo Fisher Scientific, Waltham, MA, USA) with magnetic beads and DynaMag™-2 Magnet rack (Thermo Fisher Scientific, Waltham, MA, USA). To verify the concentration, purity, and quality of extracted nucleic acids, a NanoPhotometer^®^ N60 spectrophotometer (Implen, Munich, Germany) was used and their integrity was checked by electrophoresis in 1% agarose gel.

After the extraction stage, the probiotic and yeast viabilities were evaluated by PMA-qPCR using the ABI real-time system 7500™ thermocycler (Thermo Fisher Scientific, Waltham, MA, USA). Amplification reactions were conducted with the addition of 12.5 µL of 2X Power SYBR^®^GreenPCR MasterMix (Applied Biosystems™), 5 µL of the DNA sample, and ultrapure water q.s. to complete 25 µL. The primers used in the analysis are described in Table 3 and all reactions were performed in triplicate. To establish the number of viable cells, the threshold cycle (Ct) of each sample was compared with standard curves [20].

### 2.7. Physicochemical Analysis

The pH and total acidity analyses were performed at the collection points described in Table 2. The pH was measured through a Orion Three Stars equipment (Thermo Fisher Scientific, Waltham, MA, USA) using a penetration electrode, model 2A04-GF (Analyzer), in triplicate. The total acidity was determined according to the method proposed by the Instituto Adolfo Lutz [23] for alcoholic beverages, through acid neutralization titration in a standardized alkali solution, using a pHmeter up to the equivalence point. Data were expressed in grams of acetic acid per 100 mL of sample.

The soluble solids content (°Brix) was measured using a refractometer (HSR-500, Atago, Japan) from a drop of each formulation at each stage of beer production (Table 2). Specific gravity (SG), as well as alcohol content by volume (%ABV), were calculated from the °Brix value of wort before and after fermentation on the BeerSmith^TM^ software; this used the corrected original gravity (OG) value, which corresponds to the post-brewing wort, and the final gravity (FG) value, which is measured after SG value stabilization at the end of the alcoholic fermentation.

### 2.8. Metabolomic and Flavor Component Analysis by Headspace Solid-Phase Microextraction (HS-SPME) Coupled with Gas Chromatography–Mass Spectrometry (GC–MS)

Metabolomic analyses were performed using a pool of three samples for each beer sample on days 28 and 58 in duplicate. According to Gianetti et al. [24], with some modifications, 2 g of each sample were placed in a 20 mL auto-sampler headspace glass vial, with 1 g of sodium chloride to improve extraction efficiency, after degassing in an ultrasonic bath for 5 min at 5 °C to remove CO_2_. The vials were sealed and then a PTFE/silicone septum and a magnetic screw cap were used (Macherey-Nagel, Bethlehem, PA, USA).

The beer formulations volatile compounds profile was evaluated by headspace solid-phase microextraction (HS-SPME) coupled with gas chromatography–mass spectrometry (GC–MS) analysis. A Nexis GC-2030 gas chromatograph equipped with a Split/Splitless Injector (SPL) (SPME glass liner, 0.75 mm ID) and an AOC-6000 Plus autosampler (Shimadzu, Nakagyo-ku, Kyoto, Japan) was used.

A GCMS-QP2020 NX Quadrupole mass spectrometer was used to interface the GC (Shimadzu, Kyoto, Japan) and DVB/CAR/PDMS (divinylbenzene/carboxen/polydimethylsiloxane) Smart Fiber (80 μm) from Shimadzu (Kyoto, Japan) used for HS-SPME extraction. Following the manufacturer’s instructions, this fiber was preconditioned and two blank injections at 240 °C were then performed before the analysis. GC analysis was performed under a constant helium gas flow (1 mL/min) and on a PEG (polyethylene glycol) capillary column (HP-INNOWAX, 30 m, 0.25 mm i.d., 0.15 μm) supplied by Shimadzu. 

The injection was conducted with a split ratio of 10 set in splitless mode with a splitless time of 1 min. The samples were incubated for 5 min at 60 °C; then, at the same temperature for 10 min, the fiber was exposed and then inserted into the GC injector port after the extraction time, and the volatile compounds were thermally desorbed for 3 min at 230 °C. The following temperature program was applied: 40 °C for 3 min, 5 °C/min to 150 °C, 15 °C/min to 200 °C, and 200 °C for 2 min in the GC oven.

Mass spectrometry detection was performed at 70 eV by operating in the full-scan acquisition mode in the 40–350 amu range under electron impact (EI) ionization. The ion source and the temperature of the transfer line were maintained at 250 °C. For data acquisition, the total ion current (TIC) mode was performed. The mass spectra in the NIST MS database library 2020 (National Institute of Standards and Technology, Gaithersburg, MD, USA) were used to compare the molecular fragmentation and identify the volatile compounds. An SI (similarity index) above 85 was accepted for the identification of compounds. The retention index, using a series of n-alkanes (C8–C23) as reference, was calculated to perform the uncertain identifications.

The CAS Registry Number in the Yeast Metabolome Database [25] was used for each of the compounds found; it was used to evaluate their activities during fermentation and their associations with the yeast, and to assess the flavor and aroma profile in the perflavory database (http://perflavory.com/search.php (accessed on: 30 August 2022).

### 2.9. Sensory Analysis

The sensory acceptability test was performed after 30 days of storage under refrigeration at 4 °C, using a 9-point hedonic scale [26]. In addition to the general acceptability of the beverages, the purchase intention for each formulation was also evaluated. The scales used in the analysis are shown in Table 4.

The four different beer formulations were randomly offered to each participant in a way that each formulation was evaluated at least 50 times [27]. Participants in the analysis were consumers (untrained volunteer tasters) aged between 25 and 65 years, including healthy adults of both genders. Before beginning to evaluate the beer formulations, volunteers read and signed a written consent form, to ensure that they agreed to proceed with the sensory evaluation.

As inclusion criteria, the following were adopted: healthy adults; craft beers consumers. The exclusion criteria was as follows: volunteers should not be pregnant, not have an allergy or intolerance to any of the ingredients of the drink, including gluten or any other type of restriction (such as chronic illness or medical treatment with the use of drugs that may interact with the ingredients and/or or have some restriction regarding alcoholic intake); not drive vehicles for at least 60 min after the end of the analysis; not have a flu or cold; and not have been in contact with strong-smelling materials, food or cosmetics before the sensory section.

Three samples containing 50 mL of each formulation were offered to the volunteers in a monadic way, immediately after taking them from the refrigerator, in clear plastic cups randomly identified with 3 digits. Participants were instructed to drink water between the assessments of each sample offered. The sensory analysis took place after Research Ethics Committee of Faculty of Pharmaceutical Sciences at University of São Paulo approval (CAAE: 43047721.2.0000.0067).

### 2.10. Statistical Analysis

Minitab^®^ software version 21.2 (Minitab LCC, Pennsylvania, State College, PA, USA) was used for the data statistical analysis. The results for microorganisms’ viability, fermentability assay, pH, and total acidity analysis were expressed as mean ± standard deviation (SD) and were evaluated for homogeneity and normal distribution. Once the normal distribution was confirmed, the Analysis of Variance (ANOVA) was applied to compare the samples, followed by the Tukey test. For the sensory analysis, the results for acceptability and purchase intention were expressed as median ± standard deviation (SD) and the Kruskal–Wallis test was performed to evaluate significant differences between formulations. The results were considered as significantly different when *p* ≤ 0.05 (5% significance level).

## 3. Results and Discussion

### 3.1. Fermentability Assay

The fermentability assay evaluated the strain’s capacity to ferment the *S. mombin* by-product and the results of this assay are shown in Table 5. The results show that the *S. mombin* by-product had no negative or positive influence on the probiotic strain in the media with modified MRS (rich in carbohydrates), as the bacterial populations were not significantly different (*p* > 0.05) between these two media: MRS broth with phenol red (MVF) and MRS broth with phenol red and *S. mombin* by-product flour (MVF-BP). High populations were observed, reaching values of 8 log CFU/mL after 48 h of fermentation. In this sense, in a medium that provided all the nutrients necessary to improve *Lactobacillus* population, the addition of *S. mombin by-product* had no impact on bacterial growth.

However, in the assay carried out on the wort, the addition of *S. mombin* by-product impaired the growth of probiotic strains, as well as the presence of hops. Although hops are recognized as a limiting factor for bacterial growth in beer [2,4], they did not impair the probiotic growth as much as the *S. mombin* by-product did, since the probiotic viability reached 6.83 log CFU/mL in the hopped wort without the by-product. On the other hand, in the hopped wort with the *S. mombin* by-product, it was not possible to recover the probiotic microorganism after 24 h of fermentation by the counting method used.

*S. mombin* has high levels of carotenoids, phenolic compounds, and antioxidant activity. However, when it comes to its by-product, there are few studies assessing its phenolic compounds profile [8,28,29,30]. Bataglion et al. [29] evaluated the phenolic composition of *S. mombin* pulp and found a high gallic acid content (577.03 µg/g DWP), a compound that has antimicrobial and anti-carcinogenic activity in addition to a cardiovascular and neurological protective effect [11].

Regarding *S. mombin* by-product phenolic compounds, Amariz et al. [31] found a high polyphenols content (above 500 mg EAG.100 g^−1^) and antioxidant activity. Although there is a need for further studies regarding the profile of these phenolic compounds and a possible microbial activity in the *S. mombin* pulp and by-product, the fermentability assay of the present study showed that the presence of these ingredients in brewer’s wort represents limitations for the *L. paracasei* F19 fermentation and growth.

The unhopped wort without *S. mombin* by-product also showed good probiotic viability (7.78 log CFU/mL). However, hops are essential for the classification of the beverage as beer [4]. In this sense, they are necessary for brewing a sour beer.

Another important factor is the relationship between *Lactobacillus* spp. and yeasts. Some studies have shown the importance of the co-fermentation of these microorganisms in environments with low pH, such as sour beers and fermented milk [16,32]. Suharja et al. [32] compared the survival of the probiotic *L*. *rhamnosus* HN001 with and without the yeast *S. cerevisiae* EC1118 at different fermentation temperatures (30 °C and 40 °C). The authors observed that in both assays, the viability of *L*. *rhamnosus* was higher when in co-fermentation with *S. cerevisiae* after 5 weeks.

In the assay conducted, the probiotic fermentation was evaluated without the presence of yeast. The interactions between lactic acid bacteria and yeast are complex, and are not yet fully understood as they can generate negative or positive effects depending on factors such as the food matrix or strain variation [16,32]. The results of this analysis showed that the *L. paracasei* F19 strain is capable of fermenting *S. mombin* residues. However, adjustments are needed in the sour beer production process for these microorganisms to remain viable in the beverage.

### 3.2. pH and Total Acidity of Beer Samples

Brewing demanded 28 days, which included the lactic and alcoholic fermentation, maturation, and carbonation processes. The pH gradually decreased in the collected samples during the first days, which suggests acidification of the medium due to the production of organic acids by the bacteria; this was due to the addition of *S. mombin* bagasse and/or juice, which is an acidic fruit. A pronounced decrease in the pH value was observed in all formulations, followed by a stability from day 8, which corresponds to the alcoholic fermentation end and the maturation process start (Figure 2). This stabilization can also be observed regarding the viability of microorganisms, whose population remained stable after day 8.

The beer pH values ranged from 3.50 (bagasse formulation) to 3.93 (control formulation) among formulations at the end of production (T28). This showed that all formulations fit in the low pH characteristic for this style of beer, ranging from 3.0 to 3.9 [4,16]. Alcine Chan et al. [16], in their study on the survival of *Lacticaseibacillus paracasei* L26 in co-fermentation with *S. cerevisiae*, observed similar behavior for pH and acidity.

Ciosek et al. [33] also observed a similar result for pH value in their study of a sour beer produced with *S. cerevisiae* and *L. brevis*. When adding the LAB first, allowing a lactic fermentation before the alcoholic one—as in the present study—the pH showed a low pH of 3.4 after 7 days, as observed for the formulations with *S. mombin* bagasse and/or juice on day 8 of this study (FB = 3.6; FJ = 3.7; FBJ = 3.4).

The decrease in pH was followed by an increase in acidity, where both presented higher changes in the first days and a stabilization from day 8 (Figure 2 and Figure 3), the period corresponding to the end of the alcoholic fermentation carried out by the yeast. The increase in acidity and decrease in pH observed indicate the production of organic acids by the probiotic bacteria, especially lactic acid [33,34]. Thus, the probiotic microorganism was successful in fulfilling their technological role in the sour beer style: acidifying the wort and promoting its characteristic flavor [4,35].

### 3.3. Soluble Solids and Alcohol by Volume (%ABV)

The soluble solids analysis results for each formulation are shown in Figure 4 according to the production stage. The analysis of soluble solids was used as beer production control, tracking the alcoholic fermentation through °Brix value and specific gravity (SG). The stability of these values is an indication that the available sugar was converted into alcohol and yeast fermentation had ended. Gravity also has an important impact on beer flavor, as it is related to the concentration of fermentable sugars, nitrogen and unsaturated fatty acids, important components for the synthesis of esters, which plays an important role in the fruity flavor of beers [36,37].

The °Brix value dropped after lactic fermentation and the addition of *S. mombin* juice or bagasse (day 2). After the primary fermentation stage (day 2) and the addition of extra ingredients in the formulations, the secondary fermentation stage began. In this stage, the addition and fermentation of *S. cerevisiae* occurred and, after day 8, it was possible to identify a decrease in the value of °Brix to 6 for the formulation with bagasse (FB) and to 5 for the other formulations. The °Brix values were evaluated daily until reaching a constant value as an indication that the alcoholic fermentation had ended. These values remained constant for the rest of the production stages, as well as for the 30 days of storage, and are related to the specific gravity of the beverage. This is an important quality factor indicating whether the beer produced is according to the proposed style, Catharina Sour.

The specific gravity values, as well as the alcohol content (%ABV) of each formulation, are shown in Table 6. According to the Beer Judge Certification Program [7], the Catharina Sour style is classified as a light and refreshing wheat beer that has a balanced lactic sourness, low bitterness, light body, and moderate alcohol content.

Regarding gravity, both the original (OG) and the final gravity (FG) of the four beer formulations produced in this study, even though close to it, were not in the range defined for Catharina Sour Style by the Beer Judge Certification Program [7]. In this sense, the sour beers produced were lighter than those typical for the Brazilian style. This was reflected in the alcohol content, which was also lower than expected for Catharina Sour style. The bagasse formulation (FB) had the lowest alcohol by volume content, reaching 2.6 %ABV, and the others, FJ, FBJ, and FC showed the same value for alcohol content, 3.4 %ABV. In this way, all four formulations are considered as moderate alcohol content beer. In Brazil, only an alcohol content of less than or equal to 2% by volume is considered as a low-alcohol beer [38].

The low alcohol content is desirable when thinking about potential probiotic beers, both regarding promotion of probiotic bacteria survival in the specific beer [39] and to avoid possible health issues from abusive consumption of alcoholic beverages, which can culminate in health and social problems [40].

Probiotics are well known as health allies. In this sense, the formulation of a probiotic product ought to consider not only its possible consumption benefits, but also the product itself where the probiotic is been delivered. Since craft beer is commonly associated with a high alcoholic content, a concern about alcoholic ingestion is raised in the formulation of a probiotic beer.

On the other hand, some studies have discussed the alcohol health benefits when consumed in a moderated amount. De Gaetano et al. [40] discussed the effects of moderate beer consumption on health in their study, pointing out that a low to moderate beer consumption may lead to protection against cardiovascular illnesses in an healthy adult population. Quesada-Molina et al. [41] also discussed the possible benefits of beer consumption, associated with the polyphenols present in this type of fermented beverage. Several different phenolic compounds have been found in beer, mainly phenolic acids and tannins, and flavones and flavanols [42]. Regarding the phenolic content of beer, it is important to note that higher temperature treatments in pasteurization negatively affect its content [43]. To develop a beer with probiotic strains, it is extremely important that the pasteurization process does not occur. Therefore, in addition to probiotics, these beers have a higher phenolic content. 

In addition, beer also has a nutritional value. On average, a daily dose of beer (350 mL) has approximately 8 g of carbohydrates, 2.4% of calories for a 2000 kcal diet, in addition to minerals such as calcium, iron, magnesium, phosphorus, potassium, sodium, zinc, copper, manganese, and selenium [44]. As with all fermented beverages, beer can also play a role in the balance of the intestinal microbiota and may exert beneficial effects on digestive health, by improving the balance of permeability and the function of the intestinal barrier, thus preventing dysbiosis [45,46,47,48]. Therefore, a beer with a probiotic strain and a lower alcohol content could be a good alternative for those looking for a healthier and more moderate consumption of alcoholic beverages.

### 3.4. Probiotic and Yeast Populations

The probiotic bacteria viability at different stages of brewing was evaluated by the traditional pour-plate method and by the real-time PCR with PMA treatment technique. The yeast population was evaluated only by PMA-qPCR.

The *L. paracasei* F19 viability, using the pour-plate technique, are shown in Table 7. It can be observed that the formulation with the addition of bagasse (FB) remains statistically constant (*p* > 0.05) after the end of the alcoholic fermentation (T8), ranging from 3.93 log CFU/mL to 4.46 log CFU/mL. At the end of production (T28) this formulation showed a decrease of approximately 2 log in viability. However, in the analysis of 30 days of storage (T58), the probiotic microorganisms were able to recover, returning to a count of about 4 log CFU/mL. This might have happened due to injury caused by the stress process to which the probiotic bacteria are subjected during brewing, such as the low pH, which makes it difficult to count the bacterial population by the pour-plate method [20,49].

The probiotic population in the formulation with *S. mombin* juice (FJ) remained constant (*p* > 0.05) after the yeast fermentation (day 8). This observation indicated that, despite the initial decrease compared to the inoculum, the probiotic bacteria was able to survive during the fermentation and storage of beer. Although the values found were not sufficient to indicate some probiotic properties [14,50], these results demonstrate the potential of this formulation as a good option for the proposed sour beer.

The FBJ was the most harmful formulation for *L. paracasei* F19, and after the 30 days of storage their population was 2 log CFU/mL, the lowest value among all formulations for this step. The probiotic populations in the control formulation (FC) remained constant during the stages, and had a significant increase (*p* < 0.05) at the end of fermentation (day 28), reaching 7.28 log CFU/mL.

The real-time PCR showed similar results, where the control formulation (FC) had the highest population values at the end of 30 days of storage, reaching 6.85 log equivalent CFU/mL. This formulation was the only one that remained at a sufficient level to suggest a potential probiotic activity, according to the range commonly studied for probiotic indication, from 8 to 11 log CFU/mL [50]. These findings reject the initial hypothesis that the addition of *S. mombin* bagasse and/or juice could promote the viability of the probiotic bacteria added. The probiotic microorganisms’ count at each formulation by the PMA-qPCR technique are placed on Table 8.

For FB, the probiotic viability remained stable (*p* > 0.05) between the initial inoculum (T0) and day 2, after the 24 h of lactic fermentation. By day 8, the F19 population decreased to 5.95 log CFU/mL, probably due to the yeast fermentation impact on the bacterial viability. During the maturation stage (days 10 and 17), the lactic acid bacteria recovered, reaching values statistically equivalent (*p* > 0.05) to the initial inoculum. However, this value was not stable until the end of production and 30 days of storage, finishing with 3.07 log CFU/mL, a population much below that commonly studied for the indication of probiotic activity [50].

In the juice formulation, there was a significant drop on day 2 (*p* < 0.05) with F19 counts at 6.24 log equivalent CFU/mL. This value continued to drop after yeast fermentation on day 8 (5.48 CFU/mL) and after maturation on day 17 (4.75 CFU/mL), ending production with 5.50 log CFU/mL. Nevertheless, this population level was not maintained in the storage period, decreasing to 4.13 log CFU/mL for FJ.

As observed by the pour-plating technique results, FBJ, containing both the by-product and the juice, was the most harmful formulation for *L. paracasei* F19. There was a significant increase (*p* < 0.05) from time 0 to time 2 (end of lactic fermentation) in the probiotic population, but this value was not maintained after alcoholic fermentation (day 8), where the bacterial population decreased from 8.15 log CFU/mL to 2.32 log CFU/mL. At the end of the third stage of maturation (day 17), the bacterial population dropped to 1.72 log CFU/mL and recovered on day 28, ending the production at 5.77 log CFU/mL. However, as was observed in FJ, this value was not maintained during storage and decreased to 2.82 log CFU/mL.

These data, signaling a great loss in the viability of F19 in this formulation, is an indication of injury, even more than expected for the brewing environment; this may have been accentuated by the combined addition of *S. mombin* bagasse and juice. This ingredients combination was not able to improve the survival of the probiotic bacteria, probably due to the greater impact their addition can cause in the medium, in particular the higher decrease in pH, which contributes to the bacterial stress and makes their survival more difficult [4,20,39,49].

In the control formulation, the bacteria viability after alcoholic fermentation (day 8) was 7.50 log CFU/mL, equivalent (*p* > 0.05) to the initial inoculum of 7.59 log CFU/mL, which indicates that *L*. *paracasei* F19 was able to survive sufficiently for a possible probiotic potential [50] in hopped wort and in the presence of alcohol. The control formulation (FC) was the only one that managed to maintain constant values (*p* > 0.05) during the final stages of production: maturation, carbonation, and storage (days 17, 28, and 58), ending with 6.85 log CFU/mL. Therefore, FC was, once more, the formulation that presented the highest values regarding F19 viability.

In foods with probiotic microorganisms, it is essential that these microorganisms are active, viable, and present in sufficient amounts to exert beneficial effects on health. However, since there is no consensus on the ideal dose or amount for a product to be considered as probiotic, it is necessary to prove the therapeutic effects associated with the strain from clinical trials [14,51]. The Government of Canada suggests a minimum amount of 9 log CFU/mL to be considered as a probiotic product [14], but no studies were found regarding the evaluation of probiotic dose-response in alcoholic beverages, such as beers. Alcine Chan et al., Capece et al., and Dysvik et al. studied probiotic survival and feasibility on beer production and co-fermentation, and Silva et al. evaluated the probiotic survival of a sour beer after gastrointestinal digestion in mice [1,16,39,52]. These studies suggested that beers could be a vehicle for probiotic delivery under appropriate conditions.

Dysvik, et al. [39] evaluated different species of *Lactobacillus* in co-fermentation with *S. cerevisiae* in the production of sour beer, and found values of 7 log CFU/mL for *L*. *brevis* and *L*. *plantarum* viability. The authors found compatible results with those found in the present study for the control formulation, which reached 7.28 log CFU/mL (T28) and was prepared without any addition of *S. mombin* bagasse or juice. These findings are in accordance with the literature, in which it is reported that some *Lactobacillus* spp. are resilient enough to survive in hopped wort and all the other beer limitations factors [16,39,53]. However, further studies are needed regarding the use of other ingredients in beer production, and how they can influence the viability of probiotic microorganisms.

Although the first evaluation steps (day 0 and day 2) of viability were performed on the wort from the same batch and with similar characteristics (9.2 °Brix and 1.036 SG), the bacteria behaved in different ways (*p* < 0.05). For the formulations with bagasse (FB) and control (FC), after lactic fermentation (day 2) these values remained without any statistical differences (*p* > 0.05). For the FBJ formulation, an increase (*p* < 0.05) was observed from 7.33 log CFU/mL in the initial inoculum to 8.15 log CFU/mL. The juice formulation (JF) was the only one in which counts dropped after lactic fermentation compared to the initial inoculum, from 8.25 CFU/mL to 6.24 log CFU/mL (*p* < 0.05).

This consistency regarding bacterial viability in FC was observed in the two counting methods employed. These findings show that F19 was able to survive in a hopped wort, considered as one of the factors that narrow bacterial growth in beer [4]. *S. mombin* bagasse and juice were only added to the formulations after 24 h of lactic fermentation. Therefore, it is not possible to assess the impact of these ingredients on the day 2 stage.

In general, a similar bacterial behavior in the formulations was observed by both techniques used for probiotic quantification. In the formulations with added ingredients, the bacteria were more injured, dropping and recovering in some production steps, while the control formulation showed more stability.

On the other hand, in general, the quantification of the microorganism population by PMA-qPCR showed higher values compared to those in the pour-plating technique. This occurs since the qPCR technique is considered to be more sensitive than the traditional plate counting technique [54]. Similar results were found in other studies, where the bacterial population count was higher in the real-time PCR analysis than in the conventional pour-plating technique; this was especially evident when the bacteria were subjected to stress situations, as it is in the case gastrointestinal digestion simulation, where the pH is around 2.3 to 2.6 in the gastric phase [20,49].

In the present study, lactic acid bacteria were also subjected to a stress environment, with a pH value around 3 (Figure 2) and the presence of hops, a fact that may explain the difference in counts of bacterial populations between the methods applied. The viability of *S. cerevisiae* was quantified only by the PMA-qPCR technique, and the values obtained for each production step are shown in Table 9.

Yeast was added after 24 h of lactic fermentation with the initial inoculum around 6 log CFU/mL. Quantifications by PMA-qPCR showed that, after alcoholic fermentation (T8), only the FBJ formulation showed a significant increase (*p* < 0.05) in the yeast population. In the other formulations, the yeast population remained stable (*p* > 0.05). Similar results were found in the study by Alcine Chan et al. [16], where values around 6 log CFU/mL were observed for *S. cerevisiae* S-04 in co-fermentation with *L. paracasei* L26 after 10 days of production.

Overall, a consistency in the behavior of the yeast was observed in the formulations produced, where, for this beer style, the alcohol content is usually lower, up to 5.5 %ABV (Table 6) [7]. One of the ways of producing beers with reduced alcohol content is by limiting the yeasts’ growth, which may happen via the technique applied in this study, where a lactic fermentation took place before adding the yeast [34,35,55].

Dysvik, et al. [39] evaluated *S. cerevisiae* in co-fermentation with different species of *Lactobacillus* spp. and also observed a constant behavior in most of their formulations. In the presence of *L*. *plantarum*, the authors reported values of 3.3 × 10^2^ log CFU/mL after 21 days of production, therefore similar to those found in the present study after 28 days of production for FJ and FBJ.

The interaction in the co-fermentation or in a sequential fermentation of lactic acid bacteria with yeast, as in the case of the present study, might be negative or positive for both microorganisms. Some studies have stated that the nutrients produced by the yeasts may enhance LAB viability or impair them by ethanol production, while medium acidification, caused by LAB lactic acid production, may decrease or delay the cell growth of yeasts [16,32,33].

To summarize, the present study evaluated the survival of the LAB *L. paracasei* F19 and the yeast *S. cerevisiae* US-05 in the brewing and storage of four sour beer formulations with the addition of *S. mombin* juice and/or bagasse. The formulations with *S. mombin* showed the lowest F19 counts, indicating that these ingredients threatened probiotic survival. The control formulation was the only one that, at the end of 28 days of production and 30 days of storage, showed cell count in a sufficient amount for a perspective of a possible probiotic potential (6.85 log UFC/mL).

The PMA-qPCR technique was more sensitive and efficient regarding the microbial population count, since the LAB cells suffered injury in the brewing process and the low pH environment. *S. cerevisiae* cell counts showed a stable behavior during fermentation. Regarding alcohol content, a moderate alcohol content was obtained, which is desirable in a probiotic beer with health claims.

### 3.5. Sensory Evaluation

The beer formulations were analyzed for sensory acceptability and purchase intention. There was no statistical difference in preference between formulations (*p* > 0.05). The juice formulation (FJ) was classified as 7 on the hedonic scale, corresponding to the concept of “Like moderately” and the other formulations were evaluated with a score of 6, “Like slightly”. The same was observed for the purchase intention, where there was no statistical difference between the formulations, graded as 4 (“Probably would buy”) for FJ and graded 3 (“Maybe would buy”) for FB, FBJ, and FC (*p* > 0.05). Figure 5 shows the median data for the preference of formulations on a hedonic scale and for the purchase intention score on an intention scale.

The main product of the glucose fermentation of LAB is lactic acid, which contributes to the sour taste of sour beers styles [56]. This specific flavor was pointed out during the sensory evaluation and was considered as pleasant for some tasters, while causing oddness for others. This fact may have contributed to the score obtained on the scale regarding the general acceptance of the drinks.

Sensory evaluation was performed by consumers (“untrained volunteer’s panel”). Therefore, technical aspects of the beverage produced, and the Catharina Sour style were not evaluated. In Brazil, Pilsner beer is one of the most consumed beers, and is characterized as a light and refreshing beer [57]. Thus, the Brazilian market can be a promising one for sour beers.

### 3.6. Volatile Compounds Identification

In total, 145 compounds were identified in all samples evaluated by HS-SPME/GC–MS (Appendix A). Among all compounds identified, 31 of them were similar in all formulations on days 28 and 58. 

Seven compounds were exclusive to the control formulation, on days 28 and 58: 1,2-Epoxyundecane; Propyl octanoate; Heptanoic acid; Benzaldehyde, 4-chloro-; 2-Undecanone; n-Capric acid isobutyl ester; and 3-Tridecanone. Among these compounds, according to the perflavory database, some impart an odor of coconut, cacao, and gin (Propyl octanoate); rancid, sour, cheesy and sweat (Heptanoic acid); waxy, fruity, creamy, fatty, orris and floral (2-Undecanone); and oily, sweet, brandy, apricot, fermented and cognac (n-Capric acid isobutyl ester). The compounds imparted flavors of waxy, cheesy, fruity, dirty, and fatty (Heptanoic acid), and waxy and fruity with creamy cheese-like notes (2-Undecanone). Two other compounds were found in the control formulation on day 28: 2-Furanmethanol, 5-ethenyltetrahydro-.alpha.,.alpha.,5-trimethyl-, cis- (no odor and flavor identified), and Styrene (odor: sweet, balsam, floral and plastic). Four other compounds were detected in the control formulation on day 58. However, we can highlight two of them: 2-Undecanol (odor: fresh, waxy, clean, cloth, cotton, and sarsaparilla; flavor: waxy fatty clean oily fresh fishy nut flesh tallow), and Pentadecanoic acid, 3-methylbutyl ester (odor: waxy, banana, fruity, sweet, cognac and green; flavor: waxy, fruity, banana, green, creamy and cheesy).

In the bagasse formulation, only one compound was found on days 28 and 58: 2-Octen-1-ol, 3,7-dimethyl- (no odor and flavor identified). In the formulation with bagasse, one compound was found only on day 28 (Cyclododecanol) and two on day 58 (Cyclohexane, 1-bromo-2-methyl-, and 2,2’-Isopropylidenebis [tetrahydrofuran]), all of them without odor and flavor identified. In the formulation with bagasse and juice, 5 compounds were found only in this formulation. Among them, we can highlight Copaene (odor: woody, spicy and honey), found only on day 28, and Isopentyloxyethyl acetate (odor: floral, red rose, fruity and metallic) and 3-Cyclohexene-1-ethanol,.beta.,4-dimethyl- (odor: fruity and herbal). Ethyl dl-2-hydroxycaproate; p-Mentha-1,5-dien-8-ol; and 2,6-Di-tert-butyl-4-hydroxy-4-methylcyclohexa-2,5-dien-1-one were also found only in the formulations with bagasse and juice on day 58, but without odor and flavor identified.

In the juice formulation, 3 compounds were found on days 28 and 58, two of which can be highlighted: cis-Hept-4-enol (odor: fresh, oily, fatty, creamy, vegetable, dairy, green and grassy; flavor: sweet, green, tomato, leaf, oily, tuna, creamy, vegetable, brothy, and fishy); and Dodecanal (odor: soapy, waxy, aldehydic, citrus, green, and floral; flavor: soapy, waxy, citrus, and orange mandarin). Two compounds were found only in the juice formulation on day 28: 2-Hydroxy-iso-butyrophenone, and 3,7-Cycloundecadien-1-ol, 1,5,5,8-tetramethyl-, no odor and flavor identified. Five compounds were found only in the juice formulation on day 58, and these are highlighted: Eicosane (odor: waxy); .alpha.-Cadinol (odor: herb and wood); and 2-Naphthalenemethanol, decahydro-.alpha.,.alpha.,4a-trimethyl-8-methylene-, [2R-(2.alpha.,4a.alpha.,8a.beta.)]- (odor: woody and green).

Interestingly, the formulations had the same acceptability according to sensory analysis, except for the juice formulation, for which acceptability was slightly higher. However, this cannot be attributed to a specific compound, and even with respect to a group of compounds, it would be difficult to determine; this is because, in addition to the influence on the amount of the compound, there is its interaction with other compounds to compose the taste and odor, and several compounds mask each other. More studies in this direction are needed. It is also interesting to note that the formulation that had the most varied compounds with identified odors and flavors, and in an exclusive way, was the control formulation. It is also important to note that several evaluators reported that they felt a slightly more acidic flavor in the control formulation, which could also be due to the absence of juice or bagasse and thus a more pronounced acidity. Although there were also statements claiming that they felt an acidic taste in the other formulations, this was more evident in the control formulation. The formulation with juice and bagasse was the one that had the most positive testimonials, regarding the sensation of its aroma.

## 4. Conclusions

In conclusion, the production of a potential probiotic sour beer with *L. paracasei* F19 in a sequential fermentation with *S. cerevisiae* US-05 is feasible regarding probiotic viability. However, *S. mombin* juice and/or bagasse should not be added, and other fruit juices and/or by products ought to be tested. The moderate alcohol content achieved is important for bacterial survival and to the development of a probiotic beer with health claims. Moreover, other studies must be carried out for the evaluation of LAB viability during gastrointestinal digestion and its health benefits.

## Figures and Tables

**Figure 1 foods-11-04068-f001:**
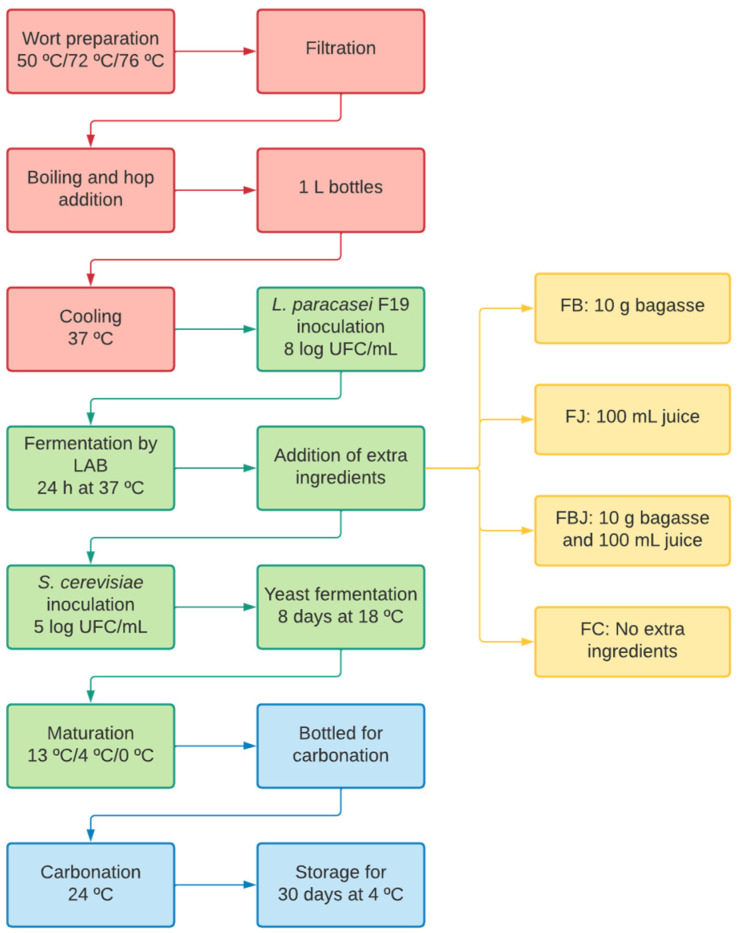
Flowchart of Sour beer production with the addition of *S. mombin* bagasse and/or juice. Wort steps in red, fermentation in green, carbonation and storage in blue, and different formulations in yellow.

**Figure 2 foods-11-04068-f002:**
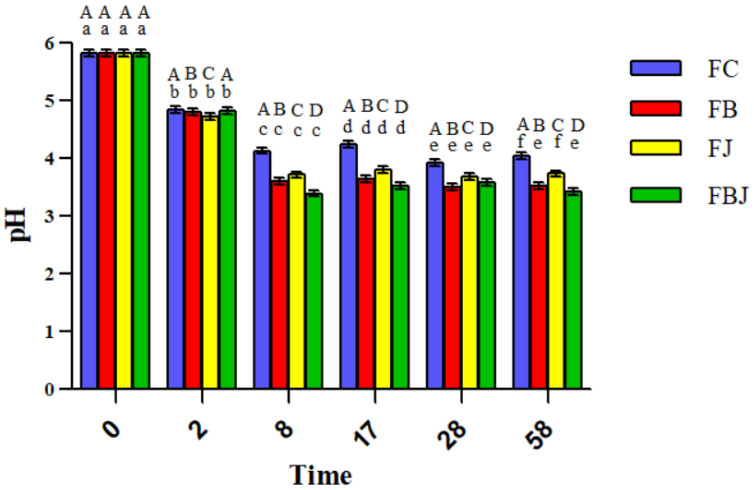
Behavior of pH of the four beer formulations. FC: control; FB: beer with bagasse; FJ: beer with juice; FBJ: beer with bagasse and juice. Day 0: beer production; day 2: end of primary fermentation/addition of bagasse and/or juice; day 8: end of alcoholic fermentation/beginning of maturation; day 17: end of the maturation stage; day 28: end of carbonation stage and beer production; day 58: after 30 days of storage at 4 °C. ^A–D^ Indicates statistical difference among columns in the same period of time (*p* < 0.05). ^a–f^ Indicates statistical difference among columns in the same formulation/color (*p* < 0.05).

**Figure 3 foods-11-04068-f003:**
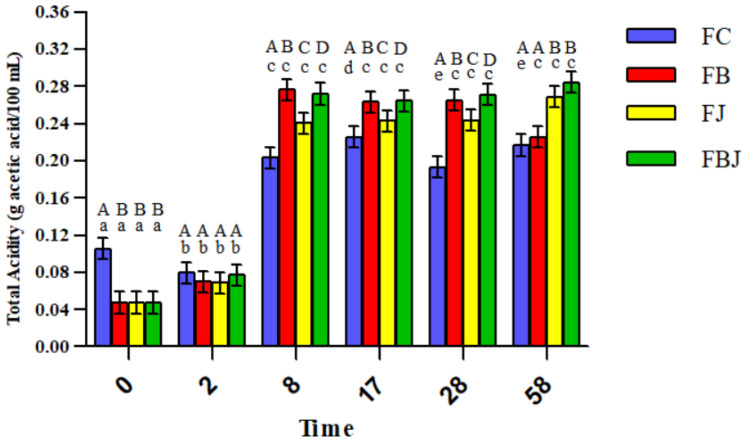
Behavior of total acidity of the four beer formulations (g acetic acid/100 mL). FC: control; FB: beer with bagasse; FJ: beer with juice; FBJ: beer with bagasse and juice. Day 0: beer production; day 2: end of primary fermentation/addition of bagasse and/or juice; day 8: end of alcoholic fermentation/beginning of maturation; day 17: end of the maturation stage; day 28: end of carbonation stage and beer production; day 58: after 30 days of storage at 4 °C. ^A–D^ Indicates statistical difference among columns in the same period of time (*p* < 0.05). ^a–e^ Indicates statistical difference among columns in the same formulation/ color (*p* < 0.05).

**Figure 4 foods-11-04068-f004:**
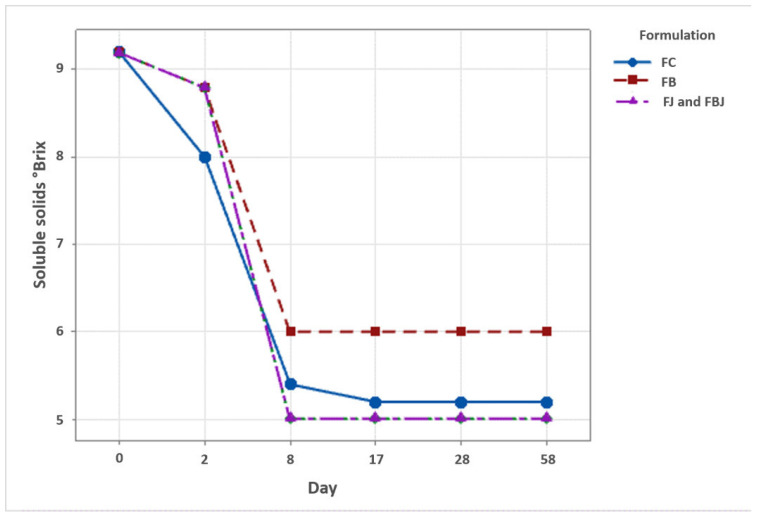
Soluble solids values of beer formulations at each production stage. FC: control; FB: beer with bagasse; FJ: beer with juice; FBJ: beer with bagasse and juice. Day 0: beer production; day 2: end of primary fermentation/addition of bagasse and/or juice; day 8: end of alcoholic fermentation/beginning of maturation; day 17: end of maturation stage; day 28: end of carbonation stage and beer production; day 58: after 30 days of storage at 4 °C.

**Figure 5 foods-11-04068-f005:**
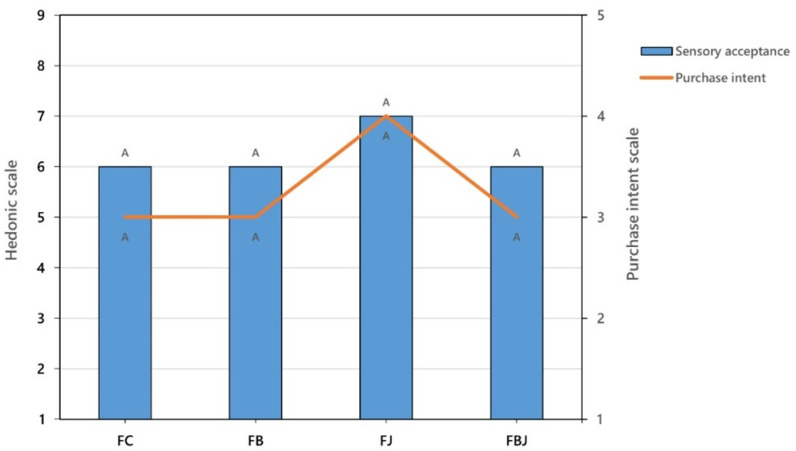
Acceptability of sensory attributes and purchase intent of the four beer formulations. FC: control; FB: beer with bagasse; FJ: beer with juice; FBJ: beer with bagasse and juice. ^A^ Different capital letters denote significant differences (*p* < 0.05). Hedonic scale: 1—Dislike extremely; 2—Dislike very much; 3—Dislike moderately; 4—Dislike slightly; 5—Neither like nor dislike; 6—Like slightly; 7—Like moderately; 8—Like very much; 9—Like extremely. Purchase intent scale: 1—Certainly wouldn’t buy; 2—Probably would not buy; 3—Maybe would buy; 4—Probably would buy; 5—Certainly would buy.

**Table 1 foods-11-04068-t001:** Formulation composition of the sour beer with the addition of bagasse and/or fruit juice.

Formulation	Composition
FC	Beer without extra ingredients (control)
FB	Beer + 10 g *S. mombin* bagasse
FJ	Beer + 100 mL *S. mombin* juice
FBJ	Beer + 10 g *S. mombin* bagasse + 100 mL *S. mombin* juice

**Table 2 foods-11-04068-t002:** Days and production steps of the formulations’ sample collection.

Day	Production Step
0	Beer production/probiotic inoculation
2	End of first fermentation/addition of extra ingredients/addition of *S. cerevisiae*
8	End of alcoholic fermentation/beginning of maturation
10	Maturation first stage
17	End of the maturation stage
28	End of carbonation/end of beer production
58	End of 30 days of storage at 4 °C

**Table 3 foods-11-04068-t003:** Primers applied to determine the viable cells equivalent numbers of *Lacticaseibacillus paracasei* subsp. *paracasei* F19 and *Saccharomyces cerevisiae* by quantitative PCR (qPCR).

Strain	Primer	Forward/ Reverse	Primer Sequence
*L. paracasei* F19 *	CRISPR2	F	5′ CGTGTGCCGATATAATGGGAACG 3′
		R	5′ CCAAAGATCATCAAGCGTGCCAT 3′
*S. cerevisiae ***	SC	F	5′ GAAAACTCCACAGTGTGTTG 3′
		R	5′ GCTTAAGTGCGCGGTCTTG 3′

* [21]; ** [22] (references with the description of primer sequences).

**Table 4 foods-11-04068-t004:** Hedonic scale for sensory acceptability and the purchase intention scale.

	Grade	Evaluation
Hedonic scale	1	Dislike extremely
	2	Dislike very much
	3	Dislike moderately
	4	Dislike slightly
	5	Neither like nor dislike
	6	Like slightly
	7	Like moderately
	8	Like very much
	9	Like extremely
Purchase intent	1	Certainly would not buy
	2	Probably wouldn’t buy
	3	Maybe would buy
	4	Probably would buy
	5	Certainly would buy

**Table 5 foods-11-04068-t005:** *L. paracasei* F19 population (log CFU/mL) in six different fermentability assay media with and without the presence of *S. mombin* by-product.

Fermentation Time	Fermentation Mediums					
	MVF	MVF-BP *	HPW	HPW-PB *	UNW	UNW-BP *
0 h	6.08 ± 0.33 ^Bab^	6.24 ± 0.07 ^Ca^	5.25 ± 0.07 ^Cc^	5.16 ± 0.06 ^c^	5.40 ± 0.08 ^Bbc^	5.48 ± 0.31 ^bc^
24 h	7.85 ± 0 ^Ab^	9.28 ± 0.10 ^Aa^	7.39 ± 0.12 ^Ac^	nd	7.54 ± 0.09 ^Abc^	nd
48 h	8.80 ± 0.28 ^Aa^	8.85 ± 0.18 ^Ba^	6.83 ± 0.18 ^Bc^	nd	7.78 ± 0.25 ^Ab^	nd

nd: not detected. * Addition of 1% of irradiated *S. mombin* by-product flour. MVF (MRS broth modified with the addition of phenol red); MVF-BP (MRS broth modified with the addition of phenol red + *S. mombin* by-product flour); HPW (hopped wort); HPW-PB (hopped wort + *S. mombin* by-product flour); UNW (unhoped wort); and UNW-BP (unhoped wort + *S. mombin* by-product flour). ^A–C^ Different superscript capital letters on the same column denote significant differences over time (*p* < 0.05). ^a–c^ Different superscript lowercase letters in the same line denote significant differences between media (*p* < 0.05).

**Table 6 foods-11-04068-t006:** Wort specific gravity and alcohol by volume (%ABV) of beer formulations and those established for Catharina Sour style beer.

Formulation	SG Wort		%ABV
	OG	FG	
FC	1.036	1.011	3.3%
FB	1.036	1.016	2.6%
FJ	1.036	1.010	3.4%
FBJ	1.036	1.010	3.4%
Catharina Sour ^1^	1.039–1.048	1.002–1.008	4.0–5.5%

^1^ Parameters for Catharina Sour beer style defined by the Beer Judge Certification Program (2021). FC: control formulation; FB: formulation with bagasse; FJ: formulation with juice; FBJ: formulation with bagasse and juice. SG: Specific gravity; OG: Original Gravity; FG: Final Gravity. %ABV: alcohol by volume percentage.

**Table 7 foods-11-04068-t007:** Populations * of *L. paracasei* F19 (log CFU/mL) in FC, FB, FJ, and FBJ formulations by the pour-plating technique during production time and storage of different beer formulations.

Formulation	Sampling Periods (Days)					
	T0	T2	T8	T17	T28	T58
FC	7.94 ± 0.05 ^Aa^	6.56 ± 0.22 ^Ca^	6.56 ± 0.22 ^Ca^	6.74 ± 0.06 ^Ca^	7.28 ± 0.10 ^Ba^	6.68 ± 0.29 ^Ca^
FB	7.69 ± 0.02 ^Ab^	6.25 ± 0.17 ^Ab^	4.45 ± 0.14 ^Bb^	3.93 ± 0.13 ^Bb^	1.50 ± 2.12 ^Cb^	4.46 ± 0.02 ^Bab^
FJ	7.94 ± 0.05 ^Aa^	6.39 ± 0.09 ^Bab^	4.33 ± 0.37 ^Cb^	4.22 ± 0.31 ^Cb^	4.54 ± 0.65 ^Cab^	4.53 ± 0.35 ^Cab^
FBJ	7.77 ± 0.03 ^Aab^	6.16 ± 0.17 ^ABb^	4.87 ± 0.72 ^BCb^	nd^Dc^	5.33 ± 1.16 ^ABa^	2.00 ± 2.83 ^CDb^

nd: not detected. * Mean ± standard deviation. FC: control formulation; FB: formulation with bagasse; FJ: formulation with juice; FBJ: formulation with bagasse and juice. ^A–D^ Different superscript capital letters on the same line denote significant differences during different sampling periods (*p* < 0.05). ^a–c^ Different superscript lowercase letters in the same column denote significant differences between formulations for a same sampling period (*p* < 0.05).

**Table 8 foods-11-04068-t008:** Populations * of *L. paracasei* F19 (log CFU/mL) in the FC, FB, FJ, and FBJ formulations by the PMA-qPCR technique during production and storage.

Formulation	Sampling Periods						
	T0	T2	T8	T10	T17	T28	T58
FC	7.59 ± 0.58 ^ABab^	9.01 ± 0.81 ^Aa^	7.50 ± 0.28 ^Ab^	2.78 ± 0.65 ^Cb^	8.43 ± 0.00 ^Aab^	7.84 ± 0.02 ^Aab^	6.85 ± 0.00 ^Aa^
FB	7.18 ± 0.08 ^Ba^	7.13 ± 0.01 ^BCa^	5.95 ± 0.03 ^Bb^	7.27 ± 0.004 ^Aa^	6.94 ± 0.08 ^Ba^	3.07 ± 0.24 ^Cc^	3.07 ± 0.24 ^Cc^
FJ	8.25 ± 0.08 ^Aa^	6.24 ± 0.03 ^Cb^	5.48 ± 0.02 ^Cc^	5.42 ± 0.01 ^Bb^	4.75 ± 0.04 ^Cd^	5.50 ± 0.02 ^Bc^	4.13 ± 0.03 ^Bb^
FBJ	7.33 ± 0.03 ^Bb^	8.15 ± 0.03 ^ABa^	2.32 ± 0 ^Dd^	6.74 ± 0.02 ^Ab^	1.72 ± 0.17 ^Dc^	5.77 ± 0.05 ^Bd^	2.82 ± 0.08 ^Cc^

* Mean ± standard deviation. FC: control formulation; FB: formulation with bagasse; FJ: formulation with juice; FBJ: formulation with bagasse and juice. T0: initial inoculum; T2: first fermentation end (F19 fermentation); T8: secondary fermentation end (alcoholic fermentation); T10: maturation first stage; T17: End of the maturation stage; T28: End of carbonation stage; T58: after 30 days of storage at 4 °C. ^A–D^ Different superscript capital letters on the same line denote significant differences during different sampling periods (*p* < 0.05). ^a–c^ Different superscript lowercase letters in the same column denote significant differences between the formulations for a same sampling period (*p* < 0.05).

**Table 9 foods-11-04068-t009:** Populations * *S. cerevisiae* US-05 (log CFU/mL) in FC (control), FB (bagasse), FJ (juice), and FBJ (bagasse and juice) formulations by PMA-qPCR during production and storage of different beer formulations.

Formulation	Sampling Periods					
	T2	T8	T10	T17	T28	T58
FC	6.01 ± 0.11 ^Ac^	5.64 ± 0.10 ^ABc^	5.71 ± 0 ^ABb^	5.23 ± 0.13 ^BCab^	4.78 ± 0.02 ^Ca^	3.00 ± 0.53 ^Db^
FB	6.19 ± 0.02 ^Bb^	6.22 ± 0.03 ^Bb^	7.60 ± 0.00 ^Aa^	5.37 ± 0.09 ^Ca^	4.50 ± 0.12 ^Da^	4.61 ± 0.04 ^Da^
FJ	6.36 ± 0.001 ^Aa^	6.30 ± 0.01 ^Aab^	5.50 ± 0.07 ^Bc^	4.94 ± 0.05 ^Cb^	3.96 ± 0.22 ^Db^	3.72 ± 0.37 ^Db^
FBJ	5.75 ± 0.01 ^Bd^	6.37 ± 0.01 ^Aa^	5.33 ± 0.04 ^Cd^	4.50 ± 0.12 ^Ec^	3.59 ± 0.03 ^Fc^	4.66 ± 0.02 ^Da^

* Mean ± standard deviation. FC: control formulation; FB: formulation with bagasse; FJ: formulation with juice; FBJ: formulation with bagasse and juice. T0: initial inoculum; T2: first fermentation end (F19 fermentation); T8: secondary fermentation end (alcoholic fermentation); T10: maturation first stage; T17: End of maturation stage; T28: End of carbonation stage; T58: after 30 days of storage at 4 °C. ^A–F^ Different superscript capital letters on the same line denote significant differences during different sampling periods (*p* < 0.05). ^a–d^ Different superscript lowercase letters in the same column denote significant differences between formulations for a same sampling period (*p* < 0.05).

## Data Availability

All related data and methods are presented in this paper. Additional inquiries should be addressed to the corresponding author.

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
