# Peer review of "Sour Beer with Lacticaseibacillus paracasei subsp. paracasei F19: Feasibility and Influence of Supplementation with Spondias mombin L. Juice and/or By-Product"

_foods, 2022, doi:10.3390/foods11244068_

Round 1
Reviewer 1 Report
In this study, a sour beer containing Spondias mombin L. juice and by-product was brewed using synergistic fermentation of Lacticaseibacillus (L.) paracasei F19 and the yeast Saccharomyces cerevisiae US-05, and the beer was subjected to sensory evaluation, physicochemical analysis and analysis of volatile components. It also examined the effect of Spondias mombin L. juice and by-product on the fermentation characteristics of Lacticaseibacillus paracasei F19. The experimental design of this study is good and has some reference value. Overall, the manuscript has merit. However, there are some issues need to be addressed, and the paper need to be revised.
1. What kind of sugar is used for “carbonation” in line 166?
2. “modified MRS broth”, It is “no carbohydrate source”in line 173, but “rich in carbohydrates” in 331. Please give a reasonable explanation.
3. Significant difference markers need to be added in Figures 2 and 3, between different fermentation days.
4. In Section 3.1, lines 328-334, only the test results are described and no main inferences are given for the test results, to guide the reader.
5. The data of physical and chemical indexes of beer is lacking, and experimental indexes such as color, turbidity, foam stability and diacetyl can be added appropriately
6. The datas from GC-Ms are good and it is hoped that further analysis using multivariate statistical methods can be performed to find out some key volatile components. For example: Projections to latent structures (PLS) or Orthogonal projections to latent structures (OPLS).
Reviewer 2 Report
The manuscript is well written, the research is well-designed. Suggestions are given below:
Line 35-36 such as fruits, herbs, and other ingredients and spices.
…such as fruits, herbs, spices and other ingredients.
Line 104 by bleaching and drying in an air circulation oven at 60 °C for 24 hours
Please provide additional explanation for bleaching (sodium hypochlorite? used concentration…)
Line 121 The recipe was designed by the BeerSmithTM software, according to the Beer Judge Certification Program (BJCP) [7] requirements for the Catharina Sour style.
Authors should consider to provide in supplementary file brewing protocol designed in BeerSmithTM software. It is not necessary since brewing is already described in manuscript but it is worth to consider. Also, has any calculator been used to adjust the mineral content of the water used for brewing? Such as https://www.brewersfriend.com/mash-chemistry-and-brewing-water-calculator/
Line 226 Please mark 5` and 3` on primer sequence in Table 3
Figure 2 (Line 391) and Figure 3 (Line 403) Authors should consider to add error bars on graphs.
Line 476, 539, FBJ (S. mombin bagasse and juice) …
Use only the abbreviation, the explanation is given when it is first mentioned.
Line 515 The L. paracasei F19 viabilities
Viability
Table 8 Line 552 and Line 554 Populations* of L. paracasei F19 (log CFU/mL) in the FC (control), FB (bagasse), FJ (juice), and FBJ (bagasse and juice); *Mean ± standard deviation. FC: control formulation; FB: formulation with bagasse; FJ: formulation with juice; FBJ: formulation
Authors should consider to change Line 552 (i.e., L. paracasei F19 viability (log CFU/mL) in the sour beer samples with and without bagasse and/or fruit juice.) It is not necessary to repeat the meaning of abbreviations twice (in the name of the table and in the description of the table content)
Same suggestion for Table 7 (Line 531)
Round 2
Reviewer 1 Report
This manuscript has been revised and improved, and it can be considered for publication.